# Observation of an unpaired photonic Dirac point

Gui-Geng Liu [1,5], Peiheng Zhou[2,5], Yihao Yang [1,3✉], Haoran Xue [1], Xin Ren[2], Xiao Lin[1], Hong-xiang Sun [4], Lei Bi[2], Yidong Chong [1,3✉] & Baile Zhang [1,3✉]

At photonic Dirac points, electromagnetic waves are governed by the same equations as two-component massless relativistic fermions. However, photonic Dirac points are known to occur in pairs in "photonic graphene" and other similar photonic crystals, which necessitates special precautions to excite only one valley state. Systems hosting unpaired photonic Dirac points are significantly harder to realize, as they require broken time-reversal symmetry. Here, we report on the observation of an unpaired Dirac point in a planar two-dimensional photonic crystal. The structure incorporates gyromagnetic materials, which break time-reversal symmetry; the unpaired Dirac point occurs when a parity-breaking parameter is fine-tuned to a topological transition between a photonic Chern insulator and a conventional photonic insulator phase. Evidence for the unpaired Dirac point is provided by transmission and field-mapping experiments, including a demonstration of strongly non-reciprocal reflection. This unpaired Dirac point may have applications in valley filters and angular selective photonic devices.

[1] Division of Physics and Applied Physics, School of Physical and Mathematical Sciences, Nanyang Technological University, 21 Nanyang Link, Singapore 637371, Singapore. [2] National Engineering Research Center of Electromagnetic Radiation Control Materials, State Key Laboratory of Electronic Thin Film and Integrated Devices, University of Electronic Science and Technology of China, 610054 Chengdu, China. [3] Centre for Disruptive Photonic Technologies, The Photonics Institute, Nanyang Technological University, 50 Nanyang Avenue, Singapore 639798, Singapore. [4] Research Center of Fluid Machinery Engineering and Technology, Faculty of Science, Jiangsu University, Zhenjiang, China. [5] These authors contributed equally: Gui-Geng Liu, Peiheng Zhou. ✉email: yang.yihao@ntu.edu.sg; yidong@ntu.edu.sg; blzhang@ntu.edu.sg

In a two-dimensional (2D) bandstructure, the linear intersection of two bands at a single point in momentum space is called a Dirac point. Near such a point, the Bloch waves are governed by the same Dirac Hamiltonian that describes a two-component massless relativistic fermion[1–3]. It is well known that the bandstructure of the 2D material graphene[1–3] possesses Dirac points at the two inequivalent corners of its hexagonal Brillouin zone (BZ), and similar pairs of Dirac points have been realized in optical lattices[4–6] as well as classical acoustic[7–9], photonic[10–16], and plasmonic[17,18] structures. In 2D systems preserving time-reversal symmetry (T), Dirac points necessarily occur in pairs[19]. The simultaneous existence of two Dirac cones, or "valleys", implies that in order to probe features of 2D Dirac particles (e.g., pseudo-diffusion[20], Klein tunneling[21–23], and Zitterbewegung[24–26]), special precautions must be taken to ensure that only one valley is ever excited. Depending on the system geometry, this is not always possible, because defects and interfaces can give rise to strong inter-valley scattering. The realization of systems hosting unpaired Dirac points would thus open up many interesting new experimental possibilities.

There are three distinct ways in which Dirac points can appear in bandstructures. Dirac points in graphene[1–3] or photonic graphene[12] belong to the first type, which occur in lattices with unbroken T. These occur in pairs and are stabilized by a symmetry of the underlying lattice. The second type of Dirac point is associated with topologically protected surface states of three-dimensional (3D) topological insulators. Weak 3D topological insulators host topologically protected surface states consisting of paired Dirac points[27]; a photonic version of this has been demonstrated recently[14]. Strong 3D topological insulators, on the other hand, have topological surface states that form unpaired Dirac points[16,28–30]; this has been observed in condensed-matter systems[30] but has never been experimentally achieved in a classical-wave system such as a photonic crystal (PhC).

The third type of Dirac point occurs at topological phase transitions between topologically trivial and nontrivial phases with broken T[31,32]. They correspond to the transition point at which the bulk bandgap closes. This is exemplified theoretically by the Haldane model of a 2D honeycomb lattice, where an unpaired Dirac point appears during a topological transition between the conventional insulator and Chern insulator phases[32]. Notably, this requires T to be broken at the transition point (the sign of the T-breaking determines which BZ corner the Dirac point occurs at). Similarly, unpaired Dirac points can also arise in Floquet systems that are driven periodically in time[31]. Floquet systems have been experimentally simulated using 3D photonic structures of coupled waveguides[33–36], and a single Dirac cone has been demonstrated to exist in the "quasienergy" spectrum (where on-axis momentum plays the role of energy)[34]. However, many phenomena associated with unpaired Dirac points, such as one-way Klein tunneling[37], are challenging to observe in that platform due to small system sizes, lack of frequency selectivity, and other technical limitations. To our knowledge, there is no previous realization of an unpaired Dirac point in the energy (rather than quasienergy) bandstructure of any 2D classical-wave system, corresponding to the closing of a bulk bandgap in a topological phase transition.

Here, we report on the experimental observation of an unpaired photonic Dirac point in a 2D PhC operating at microwave frequencies. Here T symmetry is broken by using gyromagnetic materials biased by an external magnetic field. The PhC is fine-tuned by varying the orientation of three dielectric scatterers in each unit cell, which controls the magnitude of parity symmetry (P) breaking. At a specific orientation angle, an unpaired Dirac point appears between the second and third transverse-magnetic (TM) polarized bands, at a corner of the hexagonal BZ (some of

the methods we use to probe the unpaired Dirac point will exploit the fact that it occurs at a nonzero wave vector). The unpaired Dirac point corresponds to the transition point between a Chern insulator phase (with nonzero Chern numbers) and a topologically trivial insulator phase (with zero Chern numbers). This set of features is similar to the theoretical Haldane model in the T-broken regime (nonzero Haldane flux)[32,38,39]. Transmission measurements reveal that the closing and re-opening of the band gaps occur at the orientation angle predicted by numerical calculations. Using direct field mapping, we show that an unpolarized point source excites Bloch states in a single valley, centered on one of the two inequivalent BZ corners. Finally, we demonstrate the phenomenon of non-reciprocal reflection: microwave beams incident from different directions outside the PhC are selectively reflected, depending on the availability of an unpaired Dirac point within the PhC to couple to. The realization of an unpaired photonic Dirac point in a PhC setting opens up many possibilities for exploring the behavior and applications of 2D Dirac modes, including valley filters[40] and angular selective devices[41].

## Results

**Design of a 2D PhC with an unpaired Dirac point.** The PhC, depicted in Fig. 1a, consists of a 2D triangular lattice with lattice constant $a = 17.5$ mm. Each unit cell comprises a gyromagnetic cylinder surrounded by three right-triangular dielectric pillars (see Methods for the material parameters). The PhC is placed in an air-loaded parallel-plate waveguide (see Methods and Supplementary Fig. 1; in the photograph in Fig. 1a, the top plate has been removed for visualization). The orientation of the three dielectric pillars is characterized by an angle $\theta$, which is used to modulate the strength of the in-plane P-breaking[42]. We study TM modes, which have electric fields polarized along the cylinder axis (defined as the z-axis). For $\theta = 0°$ and zero biasing magnetic field, the PhC is P and T symmetric, and its bandstructure contains two Dirac points located at the $K$ and $K'$ points (the corners of the BZ), similar to graphene. Applying a static magnetic field $B = 0.4$ Tesla along the z-axis breaks T, and hence opens a complete photonic bandgap between the second and the third TM bands, as shown numerically in Fig. 1b. Next, increasing $\theta$ widens the bandgap at $K$ while narrowing the bandgap at $K'$. At the specific value $\theta = 12.9°$, the bandgap at $K'$ closes completely to form a Dirac point at 9.01 GHz. Further increasing $\theta$, e.g. to 30°, reopens the bandgap. Figure 1c plots the variation of the bandgap widths at $K$ and $K'$ versus $\theta$, showing the formation of unpaired Dirac points at $\theta = -12.9°$ (on $K$) and $\theta = 12.9°$ (on $K'$).

The effective Hamiltonian governing the second and third TM bands near $K$ ($K'$) is

$$\hat{H} = \nu_\mathrm{D}\left(\hat{\sigma}_x\hat{\tau}_z\delta k_x + \hat{\sigma}_y\hat{\tau}_0\delta k_y\right) + \hat{\sigma}_z\left(\hat{\tau}_z m_\mathrm{T} + \hat{\tau}_0 m_\mathrm{P}\right), \qquad (1)$$

where $\nu_\mathrm{D}$ is the group velocity around the Dirac point, $\delta k_i$ is the momentum deviation from the $K$ ($K'$) point, $\hat{\sigma}_i$ and $\hat{\tau}_i$ are the Pauli matrices operating on orbital and valley subspaces, $m_\mathrm{P}$ and $m_\mathrm{T}$ are the effective mass induced by the breaking of P and T respectively. When $m_\mathrm{P}$ and $m_\mathrm{T}$ are both zero, Eq. (1) reduces to the massless Dirac Hamiltonian. Perturbatively breaking P and/or T generates a bandgap, whose magnitude is proportional to $|m_\mathrm{P} + m_\mathrm{T}|$ at $K$ and $|m_\mathrm{P} - m_\mathrm{T}|$ at $K'$. An unpaired Dirac point forms at $K$ or $K'$ when $m_\mathrm{P} = -m_\mathrm{T}$ or $m_\mathrm{P} = m_\mathrm{T}$ respectively, which correspond to orientation angles of $\theta = -12.9°$ and 12.9° in the actual PhC with a $B = 0.4$ Tesla applied magnetic field. Besides tuning $m_\mathrm{P}$ by rotating the dielectric pillars, one can also tune $m_\mathrm{T}$ by changing the magnitude of the external magnetic field (see Supplementary Fig. 2a, b for a description of how the effective masses depend on the control parameters). Controlling both the rotation angle and the external magnetic field, a

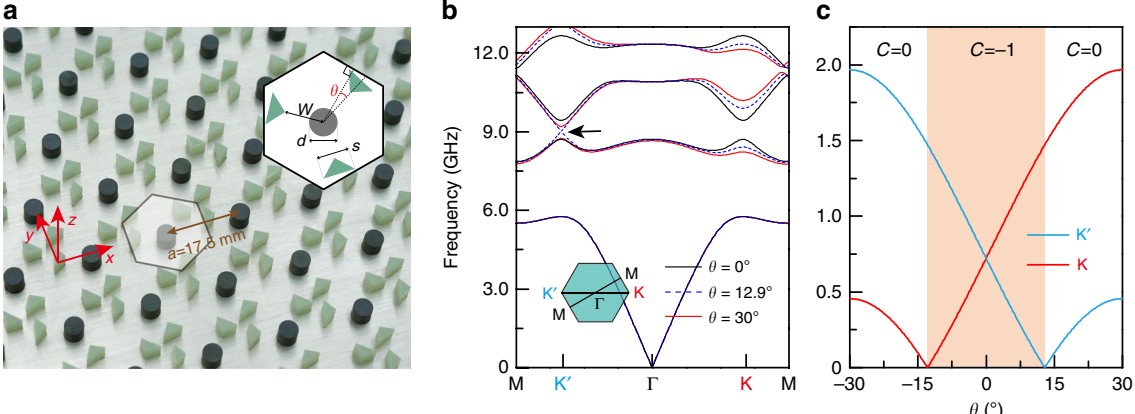

**Fig. 1 Reconfigurable gyromagnetic photonic crystals (PhCs). a** Photograph of the PhC sample in a parallel-plate waveguide. The spacing between the upper and lower aluminum plates is 4 mm. Inset: schematic of a unit cell, which consists of a gyromagnetic cylinder (dark gray circle) and three dielectric right-triangular pillars (green triangles). The gyromagnetic cylinder has diameter $d = 0.24a$, and the right-triangular pillars have hypotenuse widths $s = 0.28a$. The distance between the unit cell center and nearest triangular vertices is $w = 0.33a$. The angle $\theta$ indicates the orientation of the triangular pillars within the unit cell, relative to the three primitive vectors of the hexagonal lattice. In the depicted configuration, $\theta = 12.9°$. **b** PhC bandstructures under a 0.4 Tesla external magnetic field for $\theta = 0°$, 12.9°, and 30°. For PhC at $\theta = 12.9°$, an unpaired Dirac point appears at $K'$ (indicated by the black arrow). **c** Widths of the bandgap between the second and the third band at $K$ and $K'$, as a function of $\theta$. The bandgap at $K'$ ($K$) closes when $\theta = 12.9°$ ($\theta = -12.9°$). For $|\theta| < 12.9°$, highlighted in light orange, the bandstructure has a topologically nontrivial bandgap, with the first band having a numerically calculated Chern number of $C = -1$ (Supplementary Fig. 3).

Haldane-type phase diagram is obtained[32]. Three states with Chern number +1, −1, and 0 appear in the phase diagram. The unpaired Dirac point exists at the boundary between trivial and nontrivial states (Supplementary Figs. 2 and 3).

**Gapless bulk state and unpaired Dirac point**. We measured the bulk and edge transmissions to characterize the bandgaps and topological properties of the PhC with different orientations of dielectric pillars, with a $z$-oriented static magnetic field of $B = 0.4$ Tesla. As shown in Fig. 2a, d, g, three different domain walls are constructed between a zigzag aluminum cladding layer (acting as a trivial bandgap material) and the PhC with $\theta = 0°$, 12.9°, and 30°, respectively. We measure the bulk and edge transmissions with a vector network analyzer (Supplementary Fig. 1). The edge transmissions $S_{21}/S_{12}$ are measured when the exciting dipole antenna is placed at Port 1/Port 2, with the detecting dipole antenna at Port 2/Port 1. The bulk transmissions are measured with both source and receiver antennas placed inside the PhC, 18 cm away from each other. Figure 2a, d, g show the simulated field distributions when the point source is at Port 1 at 9.0 GHz, for the three cases.

The measured edge and bulk transmissions are plotted in Fig. 2b,e,h. For $\theta = 0°$, we find that $|S_{21}|$ is ~30 dB larger than both $|S_{12}|$ and the bulk transmission in a frequency range around 9.0 GHz. This is consistent with the existence of a topologically nontrivial bulk bandgap with a unidirectional edge state – i.e., a photonic analog of a Chern insulator[43]. For $\theta = 12.9°$, the edge and bulk transmissions almost overlap, and no distinct gap is observed, indicating that the bulk is gapless; this corresponds to the unpaired Dirac point (the dashed blue curve in Fig. 1b). For $\theta = 30°$, both the edge and bulk transmissions are gapped ~9.0 GHz, indicating that the PhC has a topologically trivial bandgap. These experimental results are further supported by the numerically calculated band diagrams of the domain wall structures, shown in Fig. 2c, f, i. Thus, by choosing the orientation of the triangle pillars in the experimental sample, we are able to precisely access the topological transition of the PhC (between Chern insulator and conventional insulator phases), where the unpaired Dirac point appears.

An important property of this unpaired Dirac point is that it occurs at the $K'$ point, and the bandstructure is gapped at $K$. Because of this, sources near the Dirac frequency only excite one valley. This is demonstrated by the experimental results shown in Fig. 3a. An unpolarized point source (labeled by a pink star) is placed inside the PhC ($\theta = 12.9°$ and $B = 0.4$ Tesla); the sample configuration is shown schematically in the inset to Fig. 3a. We measure the field patterns outside the PhC, which are caused by the refraction of the PhC's bulk states into the empty-waveguide region through a zigzag termination. The measured field pattern at 9.05 GHz, along with the corresponding numerically simulated results, are plotted. We observe that a single-directional beam is refracted out of the PhC into the empty-waveguide region, consistent with the fact that there is only a single valley of bulk states within the PhC. This valley is wave-matched to right-moving outgoing waves, as indicated in Fig. 3b, c. For comparison, we repeat the experiment for $\theta = 0°$ and $B = 0$ Tesla, with the results shown in Fig. 3d–f. In this case, there are two outgoing beams refracted into the empty-waveguide region (the operating frequency for this case, 9.76 GHz, lies near the frequency of the paired Dirac points).

**Non-reciprocal reflections**. Finally, we show that the unpaired Dirac point enables the phenomenon of non-reciprocal reflection. As shown in Fig. 4a, a diverging beam is incident on the upper boundary of the PhC slab ($\theta = 12.9°$ and $B = 0.4$ Tesla) from the left-upper empty region at 9.05 GHz (close to the Dirac frequency). The measured electric field pattern shows that reflection is suppressed at an angle of $\alpha = 39.1°$; this is confirmed by numerical simulations (see the left inset of Fig. 4a and Supplementary Fig. 4). This behavior is explained by matching tangential wave-vectors along the PhC boundary, as shown in Fig. 4b. The wave with incidence angle $\alpha$ (blue arrow) can couple to the single valley states at $K'$. Because the zigzag interface between the PhC and air preserves the valley degree of freedom[44], the beam incident at $\alpha$ undergoes "topologically protected refraction"[45], experiencing almost perfect transmission (see simulation in Supplementary Fig. 4). Waves at neighboring incidence angles lie outside the Dirac cone, and are therefore

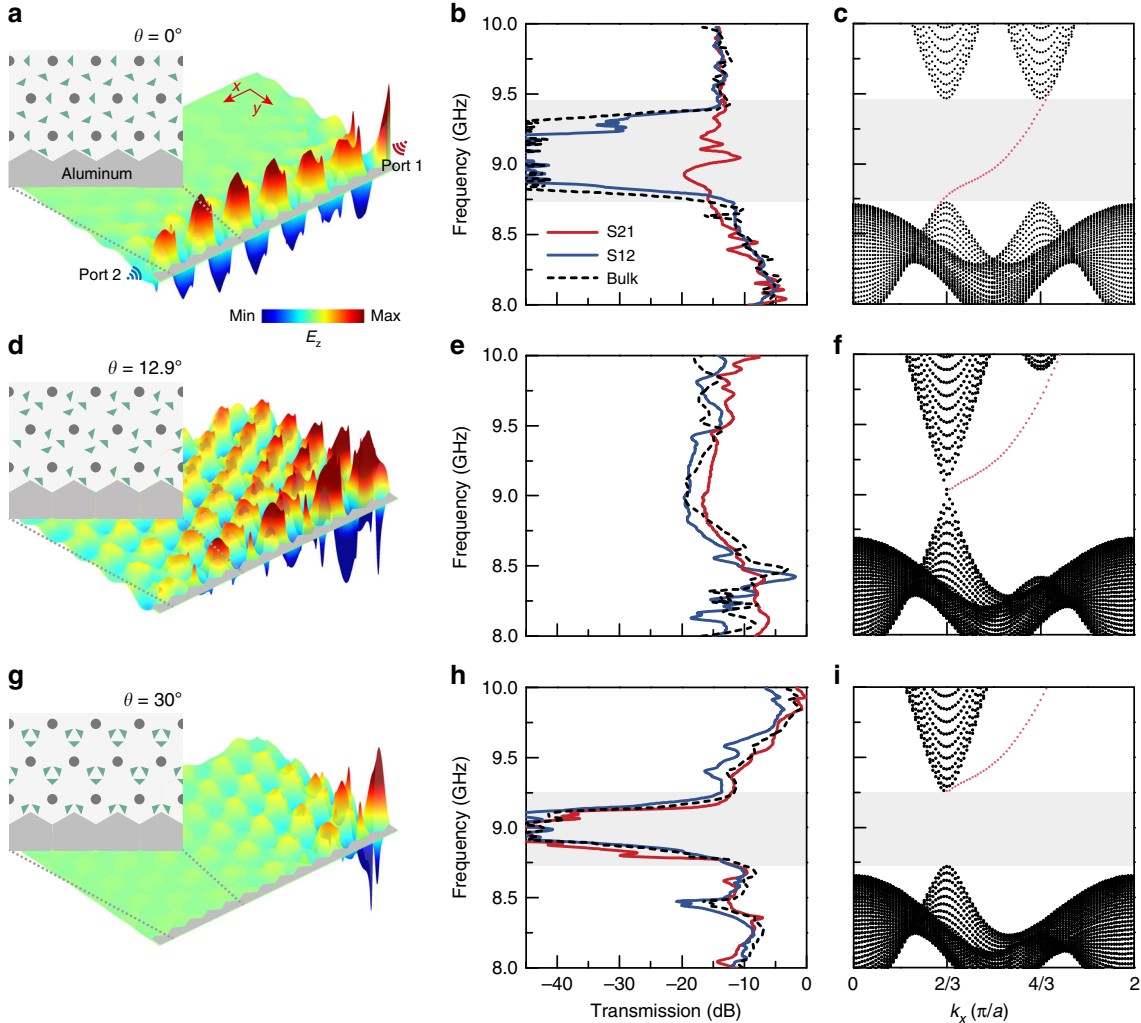

**Fig. 2 Observation of an unpaired Dirac point associated with a topological phase transition.** The orientation angles are **a–c** $\theta = 0°$ (topologically nontrivial bandgap over 8.72–9.47 GHz), **d–f** $\theta = 12.9°$ (gapless), and **g–i** $\theta = 30°$ (topologically trivial bandgap over 8.75–9.26 GHz). **a**, **d**, **g** PhC configuration, which includes a domain wall, and simulated electric field distribution excited by a 9.0-GHz source at Port 1. **b**, **e**, **h** Experimentally measured normalized bulk and edge transmissions (S-parameter magnitudes). **c**, **f**, **i** Numerically calculated band diagrams for the structures (including domain wall); black dots represent projected bulk states, and the red dots are edge states localized along the boundary. The numerical data from the third column are used to estimate the bandgap frequencies.

totally reflected. On the other hand, if a beam is incident from the right-upper region, as shown in Fig. 4c, d, strong reflection is observed even at an angle $\alpha$, as there are no valley states at K to couple into. Such a sharp contrast between left/right transmission (approaching 100% versus 0%) is difficult to been achieved in similar angular selective devices.

## Discussion

We have observed the experimental signatures of an unpaired photonic Dirac point in a 2D planar gyromagnetic PhC at microwave frequencies. The unpaired Dirac point occurs at the phase transition between a topologically nontrivial Chern insulator and a topologically trivial photonic insulator, as verified by transmission measurements. We confirm the existence of valley states at one corner of the BZ but not the other by direct field mapping of the microwaves refracting into an empty waveguide, as well as non-reciprocal reflection from the surface of the PhC. This work may provide useful guidelines for constructing unpaired Dirac points[32] at terahertz[46] or even optical frequencies[47], provided suitable gyromagnetic or gyroelectric materials can be adopted. Moreover, the availability of a practical

experimental platform hosting an unpaired Dirac point enables multiple avenues for follow-up explorations of the properties and uses of photonic Dirac modes. Two exemplary applications can be envisioned. The first is photonic valley filtering[40]: in contrast to previous time-reversal-invariant valley PhCs, which possess two opposite valleys, the current time-reversal-broken PhC has a single valley near the unpaired Dirac point, which allows for the filtering of unwanted valley signals in valley photonics. The second is non-reciprocal angular selectivity[41]: the current PhC is a perfect reflector for almost all incident angles, with an extremely narrow near-100% transmission window in a particular direction matching the unpaired Dirac point. This angular selectivity may be used to enhance the performance of non-reciprocal photonic devices.

## Methods

**Materials**. The right-triangular dielectric material used in the experiment is FR4, which has a measured relative permittivity of 4.4 and loss tangent of 0.019. The gyromagnetic material is yttrium iron garnet (YIG), a ferrite with measured relative permittivity 13.0 and dielectric loss tangent 0.0002. Its measured saturation magnetization is $M_s = 1780$ Gauss, with a ferromagnetic resonance linewidth of 35 Oe. Typically, the magnetization will decrease to a negligible value without the

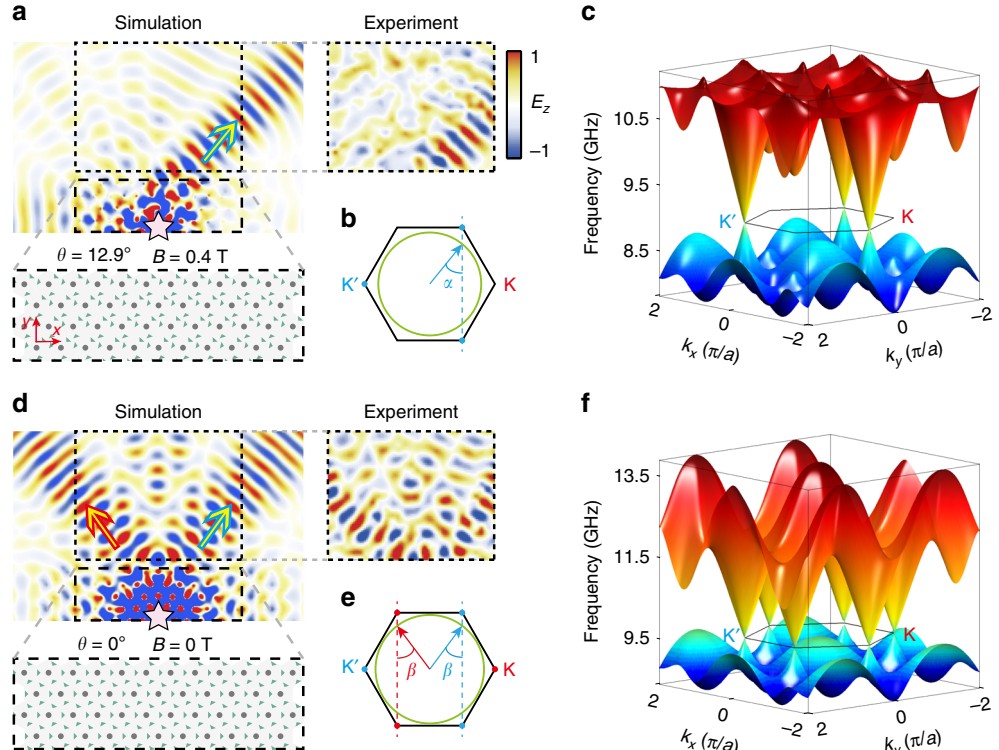

**Fig. 3 Refraction of unpaired and paired Dirac valleys into an empty waveguide. a** Experimental and numerical field maps for PhC sample with $\theta = 12.9°$ and $B = 0.4$ Tesla (whose bandstructure has an unpaired Dirac point at $K'$), with an upper zigzag boundary with an empty waveguide. The point source indicated by a pink star operates at 9.05 GHz. A single directional beam is emitted into the empty-waveguide region (blue arrow), caused by refraction from the $K'$ valley inside the PhC. **b** k-space analysis of how PhC valley modes outcouple. The three blue dots indicate the $K'$ point, and the green circle is the isofrequency surface of TM empty-waveguide modes at the Dirac frequency. The resulting outcoupling angle is $\alpha = 39.1°$, consistent with the experimental results. **c** 2D bandstructure of PhC with $\theta = 12.9°$ and $B = 0.4$ Tesla. **d** Experimental and numerical field maps for PhC sample with $\theta = 0°$ and $B = 0$ Tesla (whose bandstructure has Dirac points at $K$ and $K'$), with point source (pink star) operating at 9.76 GHz. Two directional beams are emitted into the empty-waveguide region (red and blue arrows), caused by refraction from $K$ and $K'$ valley states inside the PhC. Simulated field distribution shows mirror symmetry, while the slight distortion of the measured result is due to unavoidable experimental errors. **e** k-space analysis for the PhC with paired Dirac points, which yields an outcoupling angle $\beta = 35.8°$, consistent with experimental results. **f** 2D bandstructure of PhC with $\theta = 0°$ and $B = 0$ Tesla.

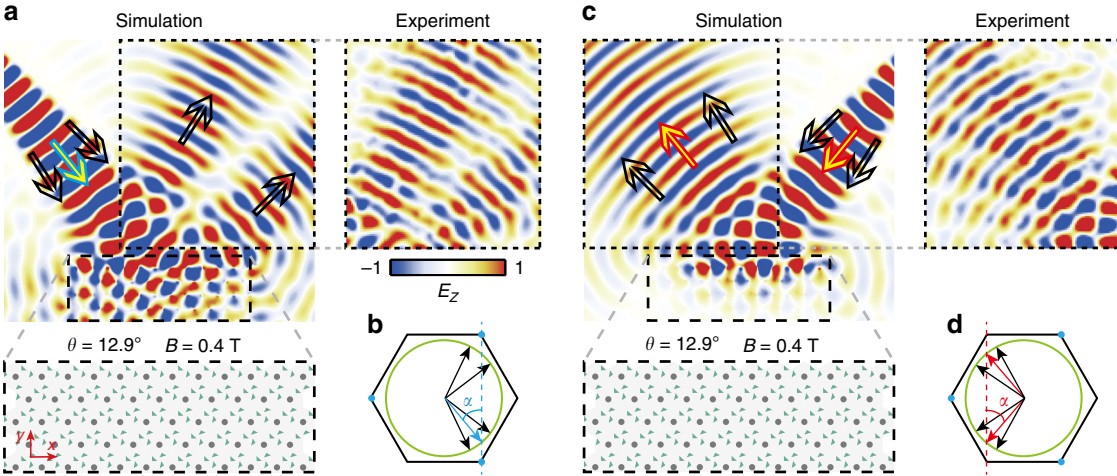

**Fig. 4 Non-reciprocal reflections due to the unpaired Dirac point. a** The incident wave from the left-upper region with the approximate angle of incidence $\alpha$ (=39.1°) can couple into the PhC slab ($\theta = 12.9°$ and $B = 0.4$ Tesla). The blue arrow represents the waves with the angle of incidence $\alpha$. The black arrows represent the waves with the angles of incidence deviated from $\alpha$ and their reflections. **b** k-space analysis on the reflection of the PhC slab of the left-upper input wave. **c** The incident wave from the right-upper region is totally reflected. The red arrows represent the wave with the angle of incidence $\alpha$ and its reflection. The black arrows represent the waves with the angles of incidence deviated from $\alpha$ and their reflections. **d** k-space analysis on the reflection of the PhC slab of the right-upper input wave.

presentce of the external magnetic field (Supplementary Fig. 1e). The relative magnetic permeability of the YIG has the form

$$\tilde{\mu} = \begin{bmatrix} \mu_r & i\kappa & 0 \\ -i\kappa & \mu_r & 0 \\ 0 & 0 & 1 \end{bmatrix}, \tag{2}$$

where $\mu_r = 1 + \frac{(\omega_0 + i\alpha\omega)\omega_m}{(\omega_0 + i\alpha\omega)^2 - \omega^2}$, $\kappa = \frac{\omega\omega_m}{(\omega_0 + i\alpha\omega)^2 - \omega^2}$, $\omega_m = \gamma M_s$, $\omega_0 = \gamma H_0$, $H_0$ is the external magnetic field, $\gamma = 1.76 \times 10^{11}$ s$^{-1}$T$^{-1}$ is the gyromagnetic ratio, $\alpha = 0.0088$ is the damping coefficient, and $\omega$ is the operating frequency[48].

**Simulation.** The dispersion relations and field patterns are simulated using the finite element software COMSOL Multiphysics. The bandstructures in Fig. 2 and Supplementary Fig. 2d–g are calculated using a supercell that is 20 cells wide, with aluminum walls are modeled as a perfect electric conductor (PEC). Both the dielectric and magnetic losses of the constituent materials are included in the simulation of Figs. 1–4 and Supplementary Fig. 2. Only the real parts of the eigenfrequency analyze are shown in the band dispersions. Since the Dirac frequency (9.01 GHz) of our demonstrate unpaired Dirac point is far away from the resonance frequency (~11.2 GHz) at 0.4 Tesla, the magnetic loss of YIG at the Dirac frequency is relative small (with loss tangent 0.018 and 0.040 for $\mu_r$ and $\kappa$, respectively). The decay length of the valley state in the PhC with unpaired Dirac point is simulated to be 26$a$, far exceeding practical structural dimensions (10$a$).

## Data availability

The data that support the plots within this paper and other findings of this study are available in the Digital Repository of NTU (DR-NTU) [https://doi.org/10.21979/N9/BFQBKH] and from the corresponding author upon request.

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

## Acknowledgements

We are very grateful to S. J. Ren and J. D. Wang from the Southwest Institute of Applied Magnetics (Mianyang, China) for providing the ferrite rods and electromagnet for the microwave experiments. This work was sponsored by the Singapore Ministry of Education under grant numbers MOE2018-T2-1-022 (S), MOE2015-T2-2-008, MOE2016-T3-1-006, and Tier 1 RG174/16 (S), and by National Key Research and Development Program of China under grant number 2016YFB1200100, and by the program of China Scholarships Council under grant number 201806075001, and by the National Natural Science Foundation of China under grant number 11774137.

## Author contributions

All authors contributed extensively to this work. G.-G.L., P.Z., H.X., and L.B. designed the structure and fabricated the sample. G.-G.L. and P.Z. performed the simulation. G.-G.L., Y.Y., P.Z., and H.X. designed the experiments. G.-G.L., P.Z., and X.R. carried out the experiments. G.-G.L., Y.Y., and P.Z. provided major theoretical analysis. G.-G.L., H.-X.S., and X.L. drew the figures. G.-G.L., Y.Y., P.Z., X.L., H.-X.S., Y.C., and B.Z. produced the manuscript. B.Z. and Y.C. supervised the project. All authors participated in discussions and reviewed the manuscript.

## Competing interests

The authors declare no competing interests.
