## [Peer Review File · Nature Communications]

Reviewers' Comments:

Reviewer #1:

Remarks to the Author:

The authors of the manuscript "Observation of an unpaired photonic Dirac point" reported an experimental effort to observe a rare phenomenon, unpaired Dirac cone. The physics is quite straightforward: by simultaneously introducing inversion symmetry and time-reversal symmetry, the Dirac cone at K and K' point can be tuned independently while when accidentally the effective mass introduced by inversion symmetry can be of the same magnitude of that by time-reversal symmetry, only one Dirac cone appears.

I notice that there are more and more publications to use classical wave systems to simulate condensed matter physics-related phenomena. However, honestly speaking, being experts in optics, we are wondering whether these novel phenomena can trigger cutting-edge technologies, which I found a lack in the current manuscript. As accidental tuning is needed, the tuning of inversion symmetry has to be exact given the gyromagnetic material used and a certain magnetic field. The 12.9-degree tuning has to be a value from a lot of try-and-error effects which makes me wonder what meaningful applications this "technology" can trigger, not to mention the gyromagnetic materials and magnetic field used limit its working frequencies to microwave only. The robustness of the phenomena is doubtful. For instance, in Fig. 3d, two Dirac cones were excited for comparison while I will expect a result with mirror symmetry while honestly, I have some conservations on the current experimental results.

Novelty is also my concern. The recipe to generate an unpaired Dirac cone is not new as Ref. 39 is an example. The experiment technique is also not new as it is the standard valley physics as similar results can be found (though have to selectively excite one Dirac point, for instance, Nature Physics 13 (4), 369 and Appl. Phys. Lett. 111, 251107)

I suggest more applications to this unpaired Dirac point can be included if resubmission is allowed. For instance, if one-way Klein tunneling is so important to be included in abstract and conclusion, the author may consider demonstrating it.

Reviewer #2:

Remarks to the Author:

In this manuscript the authors report the realization of a photonic band structure characterized by an unpaired Dirac cone in a system containing gyromagnetic materials (to break time-reversal symmetry) and parity-breaking dielectric elements. In my opinion, this work is interesting and the physics is clean, with strong arguments in support of the main claims. My only criticism concerns some aspects of the presentation. In essence, while the authors focus on the Dirac cone, the structure described in the manuscript represents the photonic correspondent of an electronic two-dimensional class A system that can be tuned between a topologically-trivial insulating phase and a Chern insulator phase. The Dirac cone corresponds to the closing of the (bulk) gap associated with the topological quantum phase transition (TQPT). In this context, I believe that a more rigorous description (in the introduction) of different scenarios associated with the presence of Dirac cones would be beneficial for the reader. One should clearly make the distinction between Dirac cones characterizing the (bulk) spectrum of topologically-trivial systems (e.g., graphene, which actually has four Dirac cones, if we take into account the spin degeneracy), Dirac cones associated with topologically-protected surface states, and Dirac cones generated by the closing of the bulk gap at TQPTs. In addition, these Dirac cones (which can be either paired or unpaired) occur at finite k-vector or at $k=0$. Several of the properties discussed in this work are specifically associated with the Dirac point being at a finite wave vector (i.e. they would not be present in a system with an unpaired Dirac point at $k=0$). Another observation concerns the effective Hamiltonian (1) (i.e. the Haldane model). The motivation behind introducing this model is not clear from the text. Can it be justified based on the numerical simulations, or does one need to consider the full phenomenology of the system (as revealed by experiment and consistent with theory)? In this context, it would be useful to have some qualitative description of the dependence of the

effective masses on the control parameters (e.g., $|m_P|$ increases with θ ; $|m_T|$ first increases with B , apparently diverges at the resonance of the effective permeability, etc.). Finally, to make clear the association of the Dirac cone with the closing of the bulk gap at the TQPT, I think that some discussion of the "phase diagram" (currently Fig. S2) in the main text would be beneficial. Also, to further strengthen the conclusions, it would be useful to show explicitly that the behavior of the system is consistent with the topology of the phase diagram. I am not implying that the mapping of the full phase boundary is necessary, but at least one additional point, e.g., corresponding to $\theta = 12.9$ and $B \approx 0.2T$, would be helpful.

Reviewer #3:

Remarks to the Author:

The manuscript by Liu et al. describes experiments on microwave scattering on gyromagnetic planar topological insulators. The main novelty is the observation of a single Dirac point, which is made possible by breaking time-reversal symmetry. Photonic topological insulators are subject to significant worldwide research efforts and the present manuscript provides valuable additions. The results are interesting and the manuscript is well-written and clearly presented. I believe the work meets the novelty criteria of Nature Communications. I am therefore inclined to recommend publication but some details are missing and should be provided before publication is warranted.

1. There is no drawing or images of the experimental setup(s). It is clear that the main body of work lies in theory and device design but nevertheless this information should be included, e.g., in the supplementary information.
2. The losses in the constituent materials are provided in the Methods but the loss of the complete structures are not included in the modelling and not addressed experimentally. A major drawback of the gyromagnetic materials is exactly the losses and since a main motivation for the interest in topological insulators is reducing losses (from backscattering), it is important to carefully assess the absorption losses introduced by the constituent materials.
3. Finally, I have some concerns about calling the structures photonic topological insulators when the experiments concern microwaves. Materials behave very differently in the photonic and microwave domains and I think this should be discussed: Is there any hope of realizing these effects in the photonic domain or are they restricted to microwaves? And why not call the structures microwave topological insulators, when this is in fact what they are?

Response Letter to Reviewers

We are grateful for the constructive comments on this manuscript (NCOMMS-19-28555) from three reviewers.

In the text below, reviewer comments are quoted in italics and followed by our detailed response. We have also revised the manuscript and the Supplementary Information based on the reviewer comments, and these updates are highlighted in blue and by a vertical red line in the left margin in those files. In the text below, the references to these updates are highlighted in a similar way.

GENERAL COMMENTS FROM 1st REVIEWER:

The authors of the manuscript "Observation of an unpaired photonic Dirac point" reported an experimental effort to observe a rare phenomenon, unpaired Dirac cone. The physics is quite straightforward: by simultaneously introducing inversion symmetry and time-reversal symmetry, the Dirac cone at K and K' point can be tuned independently while when accidentally the effective mass introduced by inversional symmetry can be of the same magnitude of that by time-reversal symmetry, only one Dirac cone appears.

Response from Authors:

We thank the reviewer for commenting that the unpaired Dirac cone we observed is “a rare phenomenon”, which has not been achieved in 2D condensed matter or classical wave systems. Although the mechanism, as originally proposed in the celebrated Haldane model [*Phys. Rev. Lett.* 61, 2015 (1988)], is well-known, the practical observation of such a phenomenon is difficult. In the following, we address the reviewer’s specific comments point-by-point.

SPECIFIC COMMENTS FROM 1st REVIEWER:

1st Reviewer -- Comment 1:

I notice that there are more and more publications to use classical wave systems to simulate condensed matter physics-related phenomena. However, honestly speaking, being experts in optics, we are wondering whether these novel phenomena can trigger cutting-edge technologies, which I found a lack in the current manuscript. As accidental tuning is needed, the tuning of inversion symmetry has to be exact given the gyromagnetic material used and a certain magnetic field. The 12.9-degree tuning has to be a value from a lot of try-and-error effects which makes me wonder what meaningful applications this "technology" can trigger; not to mention the gyromagnetic materials and magnetic field used limit its working frequencies to microwave only. The robustness of the phenomena is doubtful. For instance, in Fig. 3d, two Dirac cones were excited for comparison while I will expect a result with mirror symmetry while honestly, I have some conservations on the current experimental results.

Response from Authors:

The reviewer has raised the following four concerns: (1) whether our work, as well as many other similar publications reporting novel phenomena, can produce new technologies; (2) the setup

requires fine tuning and thus a lot of try-and-error to achieve correct parameters; (3) the gyromagnetic materials and magnetic field limit the working frequencies to microwave only; (4) Fig. 3d should exhibit mirror symmetry, but the current experimental results are not fully mirror symmetric.

We will clarify these concerns one-by-one.

Concern 1: whether our work, as well as many other similar publications reporting novel phenomena, can produce new technologies

Indeed, as noticed by the reviewer, “*there are more and more publications to use classical wave systems to simulate condensed matter physics-related phenomena.*” This is simply because of the striking fact appreciated in the photonics community that a lot of physics in condensed matter systems is also applicable to photonics. For example, applying topological physics into photonics has given birth to the emerging field of “topological photonics” [*Rev. Mod. Phys.* 91, 015006 (2019)]. It has not only produced novel photonic phenomena [e.g., the quantum-Hall-like edge states in *Nature* 461, 772 (2009)], but also triggered many unprecedented technologies such as topological lasers [e.g., *Science* 358, 636 (2017); *Science* 359, eaar4005 (2018); *Nature* 578, 246 (2020)], robust delay lines [*Nat. Phys.* 7, 907 (2011)], and robust on-chip communications [*Nat. Photon.* in press, arXiv:1904.04213].

Note that Dirac points are the “building blocks” in topological physics, playing a key role in many topological transitions. In particular, the unpaired Dirac point is the transition point in the Haldane model [*Phys. Rev. Lett.* 61, 2015 (1988)] between trivial and topological phases. The Haldane model is a foundational model for the theory of topological band insulators, but it has never been realized in any condensed matter system, due to the challenges in precise control of both the parity symmetry and the time-reversal symmetry simultaneously at atomic scales. Our demonstration of the unpaired Dirac point constitutes the first photonic realization of the Haldane-type phase transition in a photonic crystal.

Therefore, our work is of fundamental interest in both condensed matter physics and the emerging field of topological photonics. Similar to other publications in this field reporting novel phenomena, our work provides better understanding of photonic topological phases (e.g., the Haldane-type phase transition in photonics), and thus contributes at the fundamental level to the further development of topological photonic devices mentioned above.

Besides the fundamental contribution, our work serves as a proof-of-concept demonstration for two potential applications related to the unpaired Dirac point:

- 1) The first is the photonic valley filtering, which is an important component in valley photonics. In contrast to all previous time-reversal-invariant valley photonic crystals that simultaneously possess two opposite valleys, our designed photonic crystal has only one valley in the vicinity of the unpaired Dirac point, which is an excellent valley filter being able to perfectly filter out unwanted valley information.

- 2) The second is the nonreciprocal angular selectivity. As shown in Fig. 4 in main text and Fig. S4 in Supplementary Information, the demonstrated photonic crystal slab is able to fully reflect electromagnetic waves at all incident angles except a particular angle (illuminated from the left only) matched with the single Dirac point. In fact, the transmission at this particular angle can reach almost 100%, because the zigzag interface between the photonic crystal and air preserves the valley degree of freedom, forming the so-called “topologically protected refraction” [as demonstrated in our previous work in *Nat. Phys.* 14, 140 (2018)]. To the best of our knowledge, such a sharp contrast between left/right transmission reaching 100% vs 0 has never been achieved before in similar angular selective devices.

We have addressed the “topologically protected refraction” on page 6, starting from line 167, and page 6, starting from line 173 in the main text. It reads as

“Because the zigzag interface between the PhC and air preserves the valley degree of freedom⁴⁴, the beam incident at α undergoes “topologically protected refraction”⁴⁵, experiencing almost perfect transmission (see simulation in Supplementary Fig. 4).”

“Such a sharp contrast between left/right transmission (approaching 100% versus 0%) is difficult to be achieved in similar angular selective devices.”

We have added some discussion on the potential applications of the unpaired Dirac point, on page 7, starting from line 183 in the main texts. It reads as

“This work may provide useful guidelines for constructing unpaired Dirac points³² at terahertz⁴⁶ or even optical frequencies⁴⁷, provided suitable gyromagnetic or gyroelectric materials can be adopted. Moreover, the availability of a practical experimental platform hosting an unpaired Dirac point enables multiple avenues for follow-up explorations of the properties and uses of photonic Dirac modes. Two exemplary applications can be envisioned. The first is photonic valley filtering⁴⁰: in contrast to previous time-reversal-invariant valley PhCs, which possess two opposite valleys, the current time-reversal-broken PhC has a single valley near the unpaired Dirac point, which allows for the filtering of unwanted valley signals in valley photonics. The second is nonreciprocal angular selectivity⁴¹: the current PhC is a perfect reflector for almost all incident angles, with an extremely narrow near-100% transmission window in a particular direction matching the unpaired Dirac point. This angular selectivity may be used to enhance the performance of nonreciprocal photonic devices.”

We have highlighted these key points at the end of our abstract and introduction, which reads as

“This unpaired Dirac point may have applications in valley filters and angular selective photonic devices.”

“The realization of an unpaired photonic Dirac point in a PhC setting opens up many possibilities for exploring the behavior and applications of 2D Dirac modes, including valley filters⁴⁰ and angular selective devices⁴¹.”

Concern 2: the setup requires fine tuning and thus a lot of try-and-error to achieve correct parameters

We agree with the reviewer that “*accidental tuning is needed*”. Nevertheless, the unpaired Dirac point does not appear randomly. In our case, one can continuously change the magnitude of the

inversion symmetry (i.e., the rotation angle of the dielectric pillars) or the time-reversal symmetry (i.e., the external magnetic field), and the unpaired Dirac point must appear between the Chern insulator phase and normal insulator phase. Therefore, with these knowledges in mind, it is not difficult to find the unpaired Dirac point. This is very different from blind try-and-error.

We have addressed this point on page 4, starting from line 57 in the Supplementary Note 2. It read as

“In experiments, one can continuously change the magnitude of the inversion symmetry (i.e., the rotation angle of the dielectric pillars) or the time-reversal symmetry (i.e., the external magnetic field), and the unpaired Dirac point must appear between the Chern insulator phase and normal insulator phase.”

Concern 3: the gyromagnetic materials and magnetic field limit the working frequencies to microwave only

Actually gyromagnetic (or gyroelectric) materials also exist at terahertz [e.g., *Nat. Phys.* 15, 1150 (2019)], and might even be pushed to work at optical frequencies [see the topological laser with magnetic field at infrared frequencies in *Science* 358, 636 (2017)]. Therefore, although our experiments are carried out at microwave frequencies, the principle to design the photonic crystal with the unpaired Dirac point can be extended to the terahertz and optical frequencies.

We have addressed this point on page 7, starting from line 183 in the main text. It read as

“This work may provide useful guidelines for constructing unpaired Dirac points³² at terahertz⁴⁶ or even optical frequencies⁴⁷, provided suitable gyromagnetic or gyroelectric materials can be adopted.”

Concern 4: Fig. 3d should exhibit mirror symmetry, but the current experimental results are not fully mirror symmetric

Indeed, Fig. 3d should be mirror symmetric, as can be seen in the simulation (left panel of Fig. 3d). For a better comparison between simulation and experiment, we duplicate Fig. 3d as Fig. R1 in the following. The scanned field distribution in the right panel is almost mirror symmetric (we have drawn a few auxiliary lines to facilitate the comparison). The slight distortion is due to the experimental imperfection, which cannot be avoided in reality.

We have addressed this point in the caption of Fig. 3 in the main text. It reads as

“Simulated field distribution shows mirror symmetry, while the slight distortion of the measured result is due to unavoidable experimental errors.”

Figure R1 | Experimental and numerical field maps for PhC sample with $\theta=0^\circ$ and $B=0$ Tesla. The simulation results show excellent mirror symmetry and a black dotted line is indicated as the line of symmetry. The experimental result is almost mirror symmetric and a black dotted line is indicated as the line of symmetry. The double-arrowed lines are plotted for the help of visualization. The slight distortion is due to unavoidable experimental errors.

1st Reviewer -- Comment 2:

Novelty is also my concern. The recipe to generate an unpaired Dirac cone is not new as Ref. 39 is an example. The experiment technique is also not new as it is the standard valley physics as similar results can be found (though have to selectively excite one Dirac point, for instance, Nature Physics 13 (4), 369 and Appl. Phys. Lett. 111, 251107)

Response from Authors:

We agree with the reviewer that “the recipe to generate an unpaired Dirac cone” has been theoretically proposed in Ref. 39 [i.e., *Sci. Adv.* 4, eaap8802 (2018)]. In fact, this recipe has also been indicated in the famous Haldane model [*Phys. Rev. Lett.* 61, 2015 (1988)]. However, such an unpaired Dirac cone is quite difficult to be observed in photonics, as it requires time-reversal symmetry breaking. It is our work that provides the first experimental observation of an unpaired Dirac point in a planar 2D gyromagnetic photonic crystal. Hence, the theoretical discussion in Ref. 39 does not affect the novelty of our experimental work; on the contrary, the continuous theoretical efforts in the past decades show the timely value of our experimental work.

Regarding the two listed experimental papers [*Nat. Phys.* 13, 369 (2017); and *Appl. Phys. Lett.* 111, 251107 (2017)], our work is fundamentally different in both physics and experimental techniques. Both these two papers are limited to only time-reversal-invariant systems, while our system is time-reversal-symmetry broken. The breaking of time-reversal-symmetry requires magnetic field and magnetic materials (lacking in these two papers), which are exactly the reason why it is so difficult to observe the unpaired Dirac point, even though paired Dirac points have been commonly observed in photonics.

1st Reviewer -- Comment 3:

I suggest more applications to this unpaired Dirac point can be included if resubmission is allowed. For instance, if one-way Klein tunneling is so important to be included in abstract and conclusion, the author may consider demonstrating it.

Response from Authors:

The key motivation of our work is to demonstrate a novel phenomenon that is fundamental to the Haldane-type phase transition, and thus provide better understanding of topological photonic states in the emerging field of topological photonics. Following the reviewer’s suggestion, we have added discussions of potential applications such as valley filters and nonreciprocal angularly selectively devices in the revised manuscript (see our reply to Comment 1).

Regarding the one-way Klein tunneling, it is a fascinating, but separate, physical phenomenon. While both being of fundamental interest, the Klein tunneling and the Haldane model belong to different fields, and the practical realization of one-way Klein tunneling still requires further studies.

To avoid potential confusion, we have deleted “one-way Klein tunneling” in the abstract, and supplemented with application discussions on valley filters and nonreciprocal angular selective devices.

As a final remark, we note that both Reviewers #2 & #3 acknowledged the significance and novelty of our work. Reviewer #2 commented “*this work is interesting and the physics is clean, with strong arguments in support of the main claims*”. Reviewer #3 particularly praised our work that “*photonic topological insulators are subject to significant worldwide research efforts and the present manuscript provides valuable additions*” and that “*I believe the work meets the novelty criteria of Nature Communications*”.

Based on above arguments in the responses to Comment 1-3, we strongly feel that with the revisions that follow the comments by Reviewer #1 and the other two reviewers, our work now deserves publication in esteemed journals like *Nature Communications*.

GENERAL COMMENTS FROM 2nd REVIEWER:

In this manuscript the authors report the realization of a photonic band structure characterized by an unpaired Dirac cone in a system containing gyromagnetic materials (to break time-reversal symmetry) and parity-breaking dielectric elements. In my opinion, this work is interesting and the physics is clean, with strong arguments in support of the main claims.

Response from Authors:

We thank the reviewer for his/her positive comments and considering that “*in my opinion, this work is interesting and the physics is clean, with strong arguments in support of the main claims.*”. In the following, we fully address the specific comments point-by-point.

SPECIFIC COMMENTS FROM 2nd REVIEWER:

2nd Reviewer -- Comment 1:

My only criticism concerns some aspects of the presentation. In essence, while the authors focus on the Dirac cone, the structure described in the manuscript represents the photonic correspondent of an electronic two-dimensional class A system that can be tuned between a topologically-trivial insulating phase and a Chern insulator phase. The Dirac cone corresponds to the closing of the (bulk) gap associated with the topological quantum phase transition (TQPT). In this context, I believe that a more rigorous description (in the introduction) of different scenarios associated with the presence of Dirac cones would be beneficial for the reader. One should clearly make the distinction between Dirac cones characterizing the (bulk) spectrum of topologically-trivial systems (e.g., graphene, which actually has four Dirac cones, if we take into account the spin degeneracy), Dirac cones associated with topologically-protected surface states, and Dirac cones generated by the closing of the bulk gap at TQPTs. In addition, these Dirac cones (which can be either paired or unpaired) occur at finite k -vector or at $k=0$. Several of the properties discussed in this work are specifically associated with the Dirac point being at a finite wave vector (i.e. they would not be present in a system with an unpaired Dirac point at $k=0$).

Response from Authors:

We thank the reviewer for this constructive suggestion. We have added a discussion on page 2, starting from line 41, in the introduction to classify the Dirac points. It reads as

“The realization of systems hosting unpaired Dirac points would thus open up many interesting new experimental possibilities.

There are three distinct ways in which Dirac points can appear in bandstructures. Dirac points in graphene¹⁻³ or photonic graphene¹² belong to the first type, which occur in lattices with unbroken time-reversal symmetry. These occur in pairs and are stabilized by a symmetry of the underlying lattice. The second type of Dirac point is associated with topologically protected surface states of three-dimensional (3D) topological insulators. Weak 3D topological insulators host topologically protected surface states consisting of paired Dirac points²⁷; a photonic version of this has been demonstrated recently¹⁴. Strong 3D topological insulators, on the other hand, have topological surface states that form unpaired Dirac points^{16,28-30}; this has been observed in condensed-matter systems³⁰ but has never been experimentally achieved in a classical-wave system such as a photonic crystal (PhC).

The third type of Dirac point occurs at topological phase transitions between topologically trivial and nontrivial phases with broken time-reversal symmetry^{31,32}. They correspond to the transition point at which the bulk bandgap closes.”

We have added a discussion on page 3, starting from line 74 in the main text to emphasize that our realized unpaired Dirac point appear at the finite wave vector. It reads as

“(Some of the methods we use to probe the unpaired Dirac point will exploit the fact that it occurs at a nonzero wave vector)”

To avoid potential confusion, we have deleted the “enhancement of magneto-optical activity by near-zero effective refractive indices” in the introduction and conclusion since it needs an unpaired Dirac point at a zero wave vector, and supplemented with application discussions on valley filters and angular selective devices.

2nd Reviewer -- Comment 2:

Another observation concerns the effective Hamiltonian (1) (i.e. the Haldane model). The motivation behind introducing this model is not clear from the text. Can it be justified based on the numerical simulations, or does one need to consider the full phenomenology of the system (as revealed by experiment and consistent with theory)? In this context, it would be useful to have some qualitative description of the dependence of the effective masses on the control parameters (e.g., $|m_P|$ increases with θ ; $|m_T|$ first increases with B , apparently diverges at the resonance of the effective permeability, etc.).

Response from Authors:

We thank the reviewer for raising this good point. The effective Hamiltonian model (1) is a low-energy Hamiltonian model near K and K' valleys, rather than the Haldane's original tight-binding model. However, some features of our PhC are similar to the Haldane model. For example, near the unpaired Dirac point, by tuning the magnitude of parity symmetry and time-reversal symmetry breaking, the phase diagram changes between a Chern insulator phase (with nonzero Chern numbers)

and a topologically trivial insulator phase (with zero Chern numbers). This set of features is similar to the theoretical Haldane model in the T -broken regime (nonzero Haldane flux).

Besides, the effective Hamiltonian (1) can be obtained from Maxwell's equations using the plane-wave expansion (PWE) method and the $k \cdot p$ approximation in the vicinity of the Dirac points. This has been done in a previous theoretical paper, cited as Ref. 39 [i.e., *Sci. Adv.* 4, eaap8802 (2018)] in our original manuscript.

Finally, following the reviewer's suggestion, we have done the qualitative description of the dependence of the effective masses on the control parameters, as shown in Fig. R2. As the bandgap size at K (K') valley is proportional to $|m_P + m_T|$ at K and $|m_P - m_T|$ at K', we can first obtain the bandgap size at two valleys via the first-principle calculation, and then get the m_P and m_T .

We have added Fig. R2 as Fig. S2 and the qualitative description of the dependence of the effective masses on the control parameters to Supplementary Note 2, on page 4, starting from line 44, in Supplementary Information as follows:

“The relations between the effective masses induced by the breaking of P and T are plotted in Fig. S2a, b. As the bandgap size at K (K') valley is proportional to $|m_P + m_T|$ at K and $|m_P - m_T|$ at K', we can first obtain the bandgap size at two valleys by using the first-principle calculation, and then get the m_P and m_T .”

In the main text, we have highlighted this point on page 5, starting from line 116, in the main text. It reads as

“(see Supplementary Note 2 for a description of how the effective masses depend on the control parameters)”

Figure R2 | The relative (a) m_P and (b) m_T as functions of rotation angles and magnetic fields, respectively. The m_P and m_T are both normalized to $m_{P(\theta=12.9^\circ)}$ and $m_{T(B=0.4 \text{ Tesla})}$. The yellow shadows indicate the strong ferromagnetic resonance region.

2nd Reviewer -- Comment 3:

Finally, to make clear the association of the Dirac cone with the closing of the bulk gap at the TQPT, I think that some discussion of the “phase diagram” (currently Fig. S2) in the main text would be beneficial. Also, to further strengthen the conclusions, it would be useful to show explicitly that the behavior of the system is consistent with the topology of the phase diagram. I am not implying that

the mapping of the full phase boundary is necessary, but at least one additional point, e.g., corresponding to $\theta = 12.9$ and $B \approx 0.2T$, would be helpful.

Response from Authors:

We thank the reviewer for this suggestion. Following the reviewer's suggestion, we have added some discussions of the "phase diagram" in Fig. S2 on page 4, starting from line 115, in the main text. It reads as

"Besides tuning m_p by rotating the dielectric pillars, one can also tune m_T by changing the magnitude of the external magnetic field (see Supplementary Note 2 for a description of how the effective masses depend on the control parameters). Controlling both the rotation angle and the external magnetic field, a Haldane-type phase diagram is obtained³². Three states with Chern number +1, -1, and 0 appear in the phase diagram. The unpaired Dirac point exists at the boundary between trivial and nontrivial states (see Supplementary Note 2)."

Following the reviewer's suggestion, we have measured one additional point in the phase diagram. Note that the phase diagram dramatically changes around the magnetic resonance and even the Chern number flips. Because in our work we focus on the region with $B > 0.36$ Tesla, at the right of the resonance region, it is better to choose a point also located in this region. The additional measured point corresponds to $\theta = 12.9^\circ$, $B = 0.5$ Tesla. Both the measured and simulated results are shown in Fig. R3b (Fig. R3 is updated as Supplementary Figure 2 in Supplementary Information). These results manifest the existence of bulk bandgap and the absence of topological edge states. Therefore, this point has a topologically trivial phase, consistent with the phase diagram.

The simulated band dispersion when $B < 0.28$ Tesla, at the left of the resonance region are plotted in Fig. R3c-e. An unpaired Dirac point also occurs at the state $\theta = 12.9^\circ$, $B = 0.2$ Tesla, shown in Fig. R3d. However, to identify it in experiments is relatively difficult because it is not at a single-mode state at the Dirac frequency. Besides the valley state, other modes close to the center of the BZ will also be excited at the Dirac frequency.

We have added the additional experimental and simulation results in the Supplementary Note 2, on page 3 of the Supplementary Information.

Figure R3 | Haldane-type Phase diagram. (a) The Haldane-type phase diagram. The black lines denote the boundaries between the Chern insulator and the topologically trivial insulator. The three crucial states, the unpaired Dirac point ($\theta=12.9^\circ$, $B=0.4$ Tesla, red circle), the trivial insulator ($\theta=12.9^\circ$, $B=0.5$ Tesla, red square), and the paired Dirac points ($\theta=0^\circ$, $B=0$ Tesla, blue circle), are experimentally studied. For comparison, three states ($\theta=12.9^\circ$, $B=0$ Tesla, green triangle; $\theta=12.9^\circ$, $B=0.15$ Tesla, green square; $\theta=12.9^\circ$, $B=0.22$ Tesla, green star) are numerically investigated. (b) Experimental bulk / edge transmission and numerical band diagram for the PhC at the state ($\theta=12.9^\circ$, $B=0.5$ Tesla, red square). (c)-(e) Numerical band diagrams for the PhC in the comparison states. The black / red circles represent the projected bulk / edge states localized along the upper zigzag boundary. Note that domain walls are included for all the numerical studies.

GENERAL COMMENTS FROM 3rd REVIEWER:

The manuscript by Liu et al. describes experiments on microwave scattering on gyromagnetic planar topological insulators. The main novelty is the observation of a single Dirac point, which is made possible by breaking time-reversal symmetry. Photonic topological insulators are subject to significant worldwide research efforts and the present manuscript provides valuable additions. The results are interesting and the manuscript is well-written and clearly presented. I believe the work meets the novelty criteria of Nature Communications. I am therefore inclined to recommend publication but some details are missing and should be provided before publication is warranted.

Response from Authors:

We thank the reviewer for his/her positive comments and favorable recommendation. In the following, we fully address the specific comments point-by-point.

3rd Reviewer -- Comment 1:

There is no drawing or images of the experimental setup(s). It is clear that the main body of work lies in theory and device design but nevertheless this information should be included, e.g., in the supplementary information.

Response from Authors:

We thank the reviewer for this constructive suggestion.

In our experiment, the z-oriented external static magnetic field is generated by a large electromagnet; the spatial non-uniformity of the magnetic field is less than 2% across the sample. Pixeled holes with a diameter 1.8 mm and a period of 8 mm are drilled through the top aluminum plate, as shown in Fig. R4 (added as Fig. S3 in the Supplementary Information). Such holes facilitate the experimental measurement as we discussed in the main text. Note that as the holes are at deep subwavelength scale, which have negligible effect on the electromagnetic-wave mode inside the parallel waveguide around 9.0 GHz (see Fig. R4d). Two identical dipole antennas are employed as the source and receiver antennas, respectively. Both antennas are inserted into the waveguide via the holes, and connected to a vector network analyzer (R&S ZNB20) to collect the transmissions. Field distributions are measured by mapping the local field at the different holes one by one. The four sides of the samples are covered with microwave absorbers.

Figure R4 | Experimental details. (a) Photography of the sample arrangements. (b)-(c) Field-mapping setups. (d) Bandstructure of the PhC sample with $\theta=12.9^\circ$. Solid black lines represent the results simulated in 2D; red star denotes the counterpart studied in 3D. $B=0.4$ Tesla is applied for both cases.

We have added the details of experimental setups in Supplementary Note 3, on page 5. The experimental measurement methods previously presented in the Methods (Sample and experimental measurement) has also been included in Supplementary Note 3, on page 5.

3rd Reviewer -- Comment 2:

The losses in the constituent materials are provided in the Methods but the loss of the complete structures are not included in the modelling and not addressed experimentally. A major drawback of the gyromagnetic materials is exactly the losses and since a main motivation for the interest in topological insulators is reducing losses (from backscattering), it is important to carefully assess the absorption losses introduced by the constituent materials.

Response from Authors:

Thanks a lot for your careful reading and suggestions. In the microwave regime, the loss of metallic components (aluminum) is usually negligible because its penetration depth approaches to zero. The main loss of our system arises from the dielectric scatters (FR4) and gyromagnetic rods (YIG). FR4 just has dielectric loss with loss tangent of 0.019. YIG has both dielectric loss (with loss tangent 0.0002) and magnetic loss. In the original version, we did not show the magnetic loss of the YIG. In the revised version, we show the complex expression of the relative magnetic permeability of the

$$\text{YIG as } \tilde{\mu} = \begin{bmatrix} \mu_r & i\kappa & 0 \\ -i\kappa & \mu_r & 0 \\ 0 & 0 & 1 \end{bmatrix}, \text{ where } \mu_r = 1 + \frac{(\omega_0 + i\alpha\omega)\omega_m}{(\omega_0 + i\alpha\omega)^2 - \omega^2}, \quad \kappa = \frac{\omega\omega_m}{(\omega_0 + i\alpha\omega)^2 - \omega^2},$$

$\omega_m = \gamma M_s$, $\omega_0 = \gamma H_0$, H_0 is the external magnetic field, $\gamma = 1.76 \times 10^{11} \text{ s}^{-1}\text{T}^{-1}$ is the gyromagnetic ratio, $\alpha = 0.0088$ is the damping coefficient, and ω is the operating frequency. Since the Dirac frequency (9.01 GHz) of our demonstrate unpaired Dirac point is far away from the resonance frequency (~11.2 GHz) at 0.4 Tesla, the magnetic loss of YIG is relative small (with loss tangent 0.018 and 0.040 for μ_r and κ , respectively).

In the revised version, we have changed the description of the magnetic permeability of the YIG in the Methods (Materials) on page 7, starting from line 203 in the main text. It reads as

“The relative magnetic permeability of the YIG has the form

$$\tilde{\mu} = \begin{bmatrix} \mu_r & i\kappa & 0 \\ -i\kappa & \mu_r & 0 \\ 0 & 0 & 1 \end{bmatrix},$$

where $\mu_r = 1 + \frac{(\omega_0 + i\alpha\omega)\omega_m}{(\omega_0 + i\alpha\omega)^2 - \omega^2}$, $\kappa = \frac{\omega\omega_m}{(\omega_0 + i\alpha\omega)^2 - \omega^2}$, $\omega_m = \gamma M_s$, $\omega_0 = \gamma H_0$, H_0 is the

external magnetic field, $\gamma = 1.76 \times 10^{11} \text{ s}^{-1}\text{T}^{-1}$ is the gyromagnetic ratio, $\alpha = 0.0088$ is the damping coefficient, and ω is the operating frequency.”

As shown in Fig. R5 (which is the updated Fig. 1b in the main text), we have calculated the bandstructure of the PhC when the intrinsic material absorption losses are included. The real part of the resulting bandstructure has a negligible difference with the lossless case. In addition, we have carried additional transmission simulation and found that the decay length of the valley state in the

PhC with unpaired Dirac point is $26a$, far exceeding practical structural dimensions ($10a$). Based on the above analyze, we conclude that the presence of dielectric and magnetic losses of our system does not affect the existence and observation of the unpaired Dirac point.

Figure R5 | PhC band structures (real part of the eigenfrequency) under a 0.4 Tesla external magnetic field for $\theta=0^\circ$, 12.9° , and 30° , with intrinsic material absorption losses included.

To present the results with loss included in the modeling, the simulation results of the bandstructures and the field distribution shown in Figs. 1-4 in the main text have all been updated with losses included, which have a negligible difference with the lossless cases.

The effect of the losses has been presented in the Methods (Simulation) on page 8, starting from line 213 in the main text. It reads as

“Both the dielectric and magnetic losses of the constituent materials are included in the simulation of Fig. 1-4. Only the real parts of the eigenfrequency analyze are shown in the band dispersion. Since the Dirac frequency (9.01 GHz) of our demonstrate unpaired Dirac point is far away from the resonance frequency (~ 11.2 GHz) at 0.4 Tesla, the magnetic loss of YIG at the Dirac frequency is relative small (with loss tangent 0.018 and 0.040 for μ_r and κ , respectively). The simulated decay length of the valley state in the PhC with unpaired Dirac point is simulated to be $26a$, far exceeding practical structural dimensions ($10a$).”

3rd Reviewer -- Comment 3:

Finally, I have some concerns about calling the structures photonic topological insulators when the experiments concern microwaves. Materials behave very differently in the photonic and microwave domains and I think this should be discussed: Is there any hope of realizing these effects in the photonic domain or are they restricted to microwaves? And why not call the structures microwave topological insulators, when this is in fact what they are?

Response from Authors:

We agree with the reviewer that “Materials behave very differently in the photonic and microwave domains”. Indeed, the structure studied in our work is especially designed for microwave measurement. However, gyromagnetic (or gyroelectric) materials also exist at terahertz [e.g., *Nat. Phys.* 15, 1150 (2019)], and even might be pushed to work at optical frequencies [see the topological laser with magnetic field at infrared frequencies in *Science* 358, 636 (2017)]. Therefore, although

our experiments are carried out at microwave frequencies, the principle to design the photonic crystal with the unpaired Dirac point can be extended to the terahertz and optical frequencies.

We have addressed this point on page 7, starting from line 183 in the main text. It read as “This work may provide useful guidelines for constructing unpaired Dirac points³² at terahertz⁴⁶ or even optical frequencies⁴⁷, provided suitable gyromagnetic or gyroelectric materials can be adopted.”

Regarding whether our structure shall be called “microwave topological insulators” or “photonic topological insulators”, we choose the latter because the term “photonic topological insulators” has been widely accepted in the field of topological photonics to describe topologically nontrivial photonic crystals/structures over the entire electromagnetic spectrum, regardless of microwave or optical frequencies. For example, the first photonic topological insulator was demonstrated in microwave frequencies [*Nature* 461, 772 (2009)], and the first three-dimensional photonic topological insulator was designed and implemented at microwave frequencies [*Nat. Photon.* 11, 130 (2017); *Nature* 565, 622 (2019)]. This actually follows the tradition of photonic crystals, which were firstly demonstrated at microwave frequencies [*Phys. Rev. Lett.* 67, 2295 (1991)] and then later extended to other spectra including optical frequencies.

Reviewers' Comments:

Reviewer #1:

Remarks to the Author:

The authors of the manuscript made substantial efforts to address the concerns raised by all three reviewers, in which some are quite common. We all agree that this is a nice demonstration of the famous Haldane model; while the author spent quite a lot of length of the manuscript to discuss the theoretical part (which is also quite well known), the discussion on the experimental effort and further applications are still not enough.

To demonstrate a rare phenomenon is important, but I still doubt the novelty to publish in such an esteemed journal as Nature Communications with a known effect (gyromagnetic materials can break time-reversal symmetry), with standard microwave experiment technique (though reviewer 3 required more detailed experimental setup, these are standard in microwave metamaterial labs) and unavoidable tuning.

The applications mentioned are more like "I can also do this." For instance, valley filtering can easily be realized by the time-reversal-symmetry valley PhC, by selectively exciting one mode, as demonstrated in NC 10, 872 (2019). I understand that these two are of different mechanisms but to trigger some applications means that it is advanced to current technology.

Two reviewers noticed the shortcomings of gyromagnetic materials. Fabrication error is also very important and using numbers from textbooks cannot fully address the loss issue. It may not be that straightforward as the authors claimed to extend to higher frequencies as a precise geometry is needed and how to apply a magnetic field.

I am glad that the manuscript does improve a lot after considering the comments while I am still not convinced that this deserves to publish in Nature Communications.

Reviewer #2:

Remarks to the Author:

In this resubmission the authors address thoroughly the issues raised in the first round, making appropriate changes/additions to the manuscript. The revisions clarify issues such as, for example, the nature of the Dirac cone relevant to this study, the effective Hamiltonian and the dependence of different effective parameters on control parameters, the phase diagram, the potential applications associated with the realization of unpaired Dirac points, the details of the experimental setup, and the effect of dielectric and magnetic losses. I believe that these improvements of the presentation, combined with the fact that the main result has high-enough degrees of relevance and novelty, warrant the publication of the manuscript.

Reviewer #3:

Remarks to the Author:

The authors have carefully addressed my questions and concerns. Reading it again, I find it rather insightful and I recommend publication of the manuscript in its present form.

Response Letter to Reviewers

We are grateful for the constructive comments on this manuscript (NCOMMS-19-28555A) from three reviewers. In the text below, reviewer comments are quoted in italics and followed by our response.

COMMENTS FROM 1st REVIEWER:

The authors of the manuscript made substantial efforts to address the concerns raised by all three reviewers, in which some are quite common. We all agree that this is a nice demonstration of the famous Haldane model; while the author spent quite a lot of length of the manuscript to discuss the theoretical part (which is also quite well known), the discussion on the experimental effort and further applications are still not enough.

To demonstrate a rare phenomenon is important, but I still doubt the novelty to publish in such an esteemed journal as Nature Communications with a known effect (gyromagnetic materials can break time-reversal symmetry), with standard microwave experiment technique (though reviewer 3 required more detailed experimental setup, these are standard in microwave metamaterial labs) and unavoidable tuning.

The applications mentioned are more like "I can also do this." For instance, valley filtering can easily be realized by the time-reversal-symmetry valley PhC, by selectively exciting one mode, as demonstrated in NC 10, 872 (2019). I understand that these two are of different mechanisms but to trigger some applications means that it is advanced to current technology.

Two reviewers noticed the shortcomings of gyromagnetic materials. Fabrication error is also very important and using numbers from textbooks cannot fully address the loss issue. It may not be that straightforward as the authors claimed to extend to higher frequencies as a precise geometry is needed and how to apply a magnetic field.

I am glad that the manuscript does improve a lot after considering the comments while I am still not convinced that this deserves to publish in Nature Communications.

Response from Authors:

We thank the reviewer for commenting that ‘*the authors of the manuscript made substantial efforts to address the concerns raised by all three reviewers*’ and ‘*we all agree that this is a nice demonstration of the famous Haldane model.*’

The novelty of this work and the applications related to the unpaired Dirac point have been fully discussed in our last response letter. Therefore, we will not repeat these materials here. In fact, the novelty and importance of our work have been well acknowledged by Reviewer #2 who commented that ‘*the main result has high-enough degrees of relevance and novelty, warrant the publication of the manuscript,*’ and Reviewer #3 who commented that ‘*Reading it again, I find it rather insightful and I recommend publication of the manuscript in its present form.*’

As for the shortcomings of gyromagnetic (or gyroelectric) materials, we believe all previous works (*Nature* 461, 772-775 (2009); *Nat. Phys.* 15, 1150-1155 (2019); *Science* 358, 636-640 (2017)) faced

the same challenges. However, an important point of our work is that we have overcome all these challenges and have experimentally observed the unpaired Dirac point in a planar photonic crystal. Besides, though applying the current strategy directly to higher frequencies might be challenging, our work gives an important clue of how to achieve the unpaired Dirac point using photonic crystals. We also would like to mention that it was the microwave realization of the photonic Chern insulator (*Nature* 461, 772-775 (2009)) that led to the optical realization of the robust topological laser (*Science* 358, 636-640 (2017)). Therefore, we believe our work could stimulate new thinking in optics.

With the above explanation, we hope the reviewer could agree with us that our work as the first observation of an unpaired photonic Dirac point in a 2D photonic crystal deserves to publish in high-profile journals such as *Nature Communications*.

COMMENTS FROM 2nd REVIEWER:

In this resubmission the authors address thoroughly the issues raised in the first round, making appropriate changes/additions to the manuscript. The revisions clarify issues such as, for example, the nature of the Dirac cone relevant to this study, the effective Hamiltonian and the dependence of different effective parameters on control parameters, the phase diagram, the potential applications associated with the realization of unpaired Dirac points, the details of the experimental setup, and the effect of dielectric and magnetic losses. I believe that these improvements of the presentation, combined with the fact that the main result has high-enough degrees of relevance and novelty, warrant the publication of the manuscript.

Response from Authors:

We thank the reviewer for recommending the publication.

COMMENTS FROM 3rd REVIEWER:

The authors have carefully addressed my questions and concerns. Reading it again, I find it rather insightful and I recommend publication of the manuscript in its present form.

Response from Authors:

We thank the reviewer for recommending the publication.